# End-to-end learning of pharmacological assays from high-resolution microscopy images

## Abstract

Predicting the outcome of pharmacological assays based on high-resolution microscopy images of treated cells is a crucial task in drug discovery which tremendously increases discovery rates. However, end-to-end learning on these images with convolutional neural networks (CNNs) has not been ventured for this task because it has been considered infeasible and overly complex. On the largest available public dataset, we compare several state-of-the-art CNNs trained in an end-to-end fashion with models based on a cell-centric approach involving segmentation. We found that CNNs operating on full images containing hundreds of cells perform significantly better at assay prediction than networks operating on a single-cell level. Surprisingly, we could predict 29% of the 209 pharmacological assays at high predictive performance (AUC > 0.9). We compared a novel CNN architecture called "GapNet" against four competing CNN architectures and found that it performs on par with the best methods and at the same time has the lowest training time. Our results demonstrate that end-to-end learning on high-resolution imaging data is not only possible but even outperforms cell-centric and segmentation-dependent approaches. Hence, the costly cell segmentation and feature extraction steps are not necessary, in fact they even hamper predictive performance. Our work further suggests that many pharmacological assays could be replaced by high-resolution microscopy imaging together with convolutional neural networks.

## 1 Introduction

High-resolution microscopy fluorescence imaging is an increasingly important biotechnology in the field of drug discovery (Pepperkok and Ellenberg, 2006; Starkuviene and Pepperkok, 2007). This imaging biotechnology captures morphological changes induced by chemical compounds on cell cultures in a very cost- and time-efficient way (Yarrow et al., 2003) and is viewed as remedy for the current drug discovery crisis (Dorval et al., 2018). However, exploiting the wealth of information contained in those images for drug discovery is still a challenge (Simm et al., 2018) and it is important to lower entry barriers for the analysis of such data (Scheeder et al., 2018).

A typical step of conventional microscopy image analysis pipelines is to segment images into single cells and then extract cell-level feature vectors (Carpenter et al., 2006). Individual steps in the pipeline of such approaches usually require optimization of the segmentation and feature extraction procedure to the specific cell culture or assay. This is a time-consuming process in which each step potentially introduces errors and uncertainty. Furthermore, parameters for each step are typically adjusted independently of subsequent steps (Finkbeiner et al., 2015; Sommer and Gerlich, 2013). Conventional microscopy image analysis is centered on single cells and thus involves segmentation and feature extraction.

Predicting pharmacological assays on the basis of high-resolution microscopy data has first been undertaken by Simm et al. (2018) using a cell-centric approach and has led to a tremendous increase of discovery rates, 250-fold and 60-fold, in two ongoing drug discovery projects. In principle, the authors replaced the chemical features of quantitative structure-activity relationship (QSAR) models with image-derived features. However, while the authors compare several machine learning methods, they use a traditional pipeline of first segmenting individual cells and deriving features

from these. This might explain why all methods they compared performed equally well, as important information is already discarded during pre-processing.

Using cell-centric approaches implies that the local neighborhood of cells, their spatial arrangement, and relation to other cells are lost. However, Bove et al. (2017) show that the neighborhood of cells and their relative orientation plays an important role for biological processes. This information is not accessible to a model which gets features based on single cells as input. Thus, learning a model in an end-to-end fashion directly from images would be highly desirable, not only because it removes the time-consuming and computationally demanding segmentation step, but also allows learning from the cell neighborhood and spatial arrangement of cells.

Since 2012, convolutional neural networks (CNNs) have been shown in several applications to outperform conventional methods in the field of image analysis especially where large datasets are available. CNNs can have a vastly better performance than expert systems at classifying images into thousands of categories (Krizhevsky et al., 2012; Simonyan and Zisserman, 2014; He et al., 2015), recognizing traffic signs (Ciresan et al., 2011) and even pixel-accurate segmentation of images (Long et al., 2015). As CNNs learn features from images automatically they can easily be adapted to other domains and indeed have been applied to cellular image data, for example, to segment cells achieving higher accuracy than human experts (Ronneberger et al., 2015). Therefore, CNNs are a promising method for extracting biological knowledge from high-resolution imaging data for the purpose of drug design.

However, applying CNNs to high-resolution microscopy imaging data in the field of drug design poses unique challenges. State-of-the-art CNN architectures are designed for and evaluated on benchmark datasets such as ImageNet (Everingham et al., 2010; Lin et al., 2014; Russakovsky et al., 2015) in which images are of a much lower resolution than typical microscopy images. This poses a problem as increased image resolution immediately results in vastly increased memory consumption, especially for very deep architectures, making training infeasible with full resolution microscopy images while scaling or cropping such images results in loss of information. Another challenge arises from the fact that labels for microscopy images are often noisy, as typically whole images are assigned a label by experts but not all individual cells conform to this label. In Kraus et al. (2016) the authors combine methods from Multiple Instance Learning and convolutional neural networks to alleviate this problem and in the process learn to focus on correctly labeled cells. Overall, the main challenges of high-resolution image analysis of cells are handling the high resolution of the images and that typically the whole image rather than a single cell is labeled.

We use the largest public dataset, *Cell Painting* (Bray et al., 2017), which consists of high-resolution fluorescence microscopy images of cells treated with chemical compounds, to benchmark and compare convolutional neural networks against each other and against the best feature-based method. This dataset comprises 919,265 five-channel microscopy images across 30,610 tested compounds as well as single-cell features extracted using a CellProfiler (Carpenter et al., 2006) pipeline. We augmented this dataset with drug activity data for 10,574 compounds integrated from ChEMBL (Gaulton et al., 2017). However, we do not restrict activity information from ChEMBL to specific assays as that would reduce the amount of data available drastically. Therefore, the resulting labels must be viewed as noisy and learning meaningful patterns from this dataset is no easy task.

We introduce a novel network architecture that is able to cope with the characteristics of typical microscopy images, GapNet. Our architecture extracts features from full resolution images such that there is no need for scaling or cropping the input images. It then combines features from different levels of abstraction and spatial resolution before feeding the resulting features into a fully-connected classification network. We combine the features in such a way that the network can handle arbitrarily large input images.

In Section 3, we provide details on the datasets, we describe competing state-of-the-art methods previously used for similar data and the evaluation criteria, and we introduce our novel network architecture. Finally, in section 4 we present and discuss results.

## 2 DATASET

To assess our method we use a dataset released by Bray et al. (2017) which we refer to as *Cell-painting Dataset*. This dataset contains 919,265 five-channel microscopy images (using the U2OS

cell-line) across 30,610 tested compounds as well as single-cell features extracted using a CellProfiler (Carpenter et al., 2006) pipeline. For our comparison we used the most recent version of the pre-computed features, where for each image 1,783 features were extracted.

The fact that the cells in each image were treated with a specific chemical compound allows to automatically obtain labels for this dataset without the need of labeling by a human expert. We identify each chemical compound in the large bioactivity database ChEMBL (Gaulton et al., 2017). This database provides outcomes of biochemical tests, called *assays*, such as a compound's ability to inhibit a certain receptor or whether a compound exhibits a certain toxic effect. In this way, we obtained labels for this dataset as drug activity data for 10,574 compounds across 209 assays. Consequently, we created a large dataset of high-resolution images together with chemical activity data.

Concretely, our computational pipeline for labeling the images comprises the following steps: We first convert the SMILES representation of compounds into the InChiKey (Heller et al., 2013) format. We then query the ChEMBL database for compounds matching these InChiKeys resulting in 11,585 hits. For these compounds we extract two values from the database, namely pChEMBL, a numerical value on log scale indicating bioactivity, and *activity comment*, where the researcher or lab technician creating the entry on ChEMBL marked a compound as either active or inactive.

We obtain the first part of our label matrix by extracting all measurements with a pChEMBL value between 4 and 10 for IC50 (inhibitory effect) or EC50 (stimulatory effect). However, due to the high amount of noise in these labels, we aim at binary prediction tasks (active/inactive) and therefore threshold this matrix similar to other works on assay or target prediction (Mayr et al., 2018). We apply not only one but three increasing thresholds, namely 5.5, 6.5 and 7.5. As the pChEMBL value is on a log scale these thresholds represent increasingly confident activity indicators. Applying all three thresholds allows us to obtain more labeling information, hence we concatenate the three resulting matrices along the assay dimension. This means that an assay can occur multiple times in the final label matrix but at different thresholds.

The second part we obtain by extracting all compounds with a valid *activity comment*. We only allow a defined set of comments (such as "active" or "inactive" with slight variations in spelling and casing). We combine the results of both the thresholded pChEMBL values as well as the *activity comment* along the assay dimension and filter only for those assays, or in case of the first part assay/threshold combinations, where at least 10 active and 10 inactive compounds are present and remove compounds without any measurements in the remaining assays.

The final label matrix consists of 10,574 rows corresponding to compounds, 209 columns corresponding to bioactivity assays with 0.87% positive labels (active), 1.64% negative labels (inactive) and 97.49% missing labels (NA). Upon publication we will release our label matrix containing bioactivity data for the Cellpainting Dataset, for which images and pre-computed features are publicly available.

From the Cellpainting Assay dataset we extract images corresponding to compounds for which we have activity information in our label matrix. As the dataset contains multiple screens per compound we have several images per row in the label matrix. Furthermore, each screen is comprised of six adjacent images, called *views*, with a resolution of 692×520 and 5 channels, each channel corresponding to a stain used for the microscopy screen. Since individual views already contain a large number of cells we do not combine these images to obtain one large image per screen but rather use each view image individually for training and only combine the network outputs by averaging predictions. Figure 1 shows examples of images from the assay "Gametocytocidal compounds screen" labeled as (a) active and (b) inactive (for illustrative purposes, each image is a full screen image comprised of six views).

The final dataset consists of 284,035 view images which we split into training-, validation- and test set making sure that multiple images from the same sample are in the same fold. We used 70% of compounds for training (corresponding to 198,609 images), 10% as a validation set (28,632 images) and the remaining 20% (56,794 images) were held out for testing the final performance.

Due to the sparseness of the label matrix, the majority of output units for a given sample should not receive an error signal. Therefore, the loss for all output units for unlabeled assays for a given

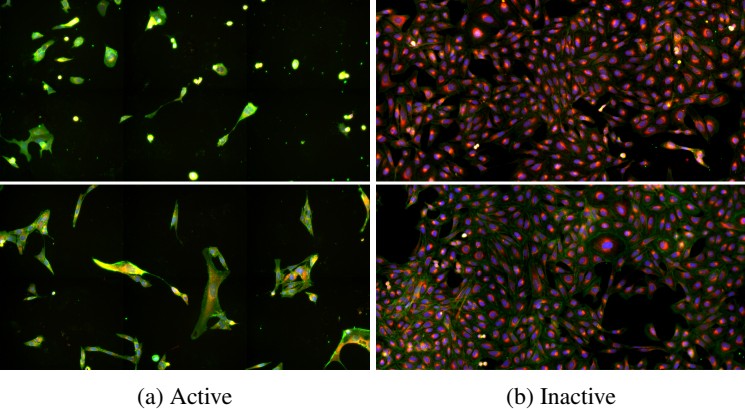

(a) Active           (b) Inactive

Figure 1: Illustrative examples from the assay "Gametocytocidal compounds screen" that are (a) labeled as active, and (b) labeled as inactive. Cells treated with *active* compounds are decreased in number and show a distinct morphology, which is clearly visible on the images. However, cell segmentation and feature extraction can be hampered because of the strong morphological changes. The samples shown here are reconstructed by combining six view images to one image.

sample are masked by multiplying it with zero before performing back-propagation to update the parameters of the network during training.

## 3 METHODS

We compare seven network architectures that were suggested for or that we could adapt to this task. For all compared methods, we manually optimized its most important hyperparameters, such as learning rate, on a validation set. We consider this setting as a multi-task problem of 209 binary prediction tasks, therefore all networks comprise 209 output units with sigmoid activations. The batch size was chosen such that the video memory of an Nvidia GTX 1080 TI was fully utilized during training.

**Convolutional Multiple Instance Learning (MIL-Net)**   In Kraus et al. (2016) the authors introduce a CNN designed specifically for microscopy data with a focus on the problem of noisy labels, i.e. that microscopy images not only contain cells of the target or labeled class but also outliers. The authors propose to tackle this problem with *multiple instance learning* (MIL), where cells belonging to the class label of an image are identified automatically while the influence of other cells on the result of the model is down-weighted by using a special pooling function called *noisyAND*. The authors implement their model using a fully convolutional approach (FCN) allowing them to train on full images with noisy labels and apply this model to images of single-cell crops. We used a learning rate of 0.01, SGD optimizer with momentum of 0.9, L2 weight decay of 0.0001 and a batch size of 64. The model specific parameter $a$ for the *noisyAND* pooling function was set to 10 as suggested by the authors.

**Multi-scale Convolutional Neural Network (M-CNN)**   The approach of Godinez et al. (2017) processes the input at several different resolutions simultaneously and fuses the resulting feature maps late in the network. Specifically the input is processed at seven scales, from original resolution to downscaled by a factor of 64. Then, the intermediate features are pooled to be of equal size and concatenated before a final convolutional layer with $1 \times 1$ kernel to combine them. This architecture was designed specifically for phenotype prediction directly from microscopy images. We used a learning rate of 0.001, SGD optimizer with momentum of 0.9, L2 weight decay of 0.0005 and a batch size of 100.

**ResNet**   The work of He et al. (2015) enabled training of very deep networks with hundreds of layers. This is enabled by *residual connections*, which are identity connections bypassing several convolutional layers allowing gradients to flow unencumbered through the network. Networks based

on this architecture are still state-of-the-art today. We use the variant ResNet-101 for our comparisons. We used a learning rate of 0.001, SGD optimizer with momentum of 0.9, L2 weight decay of 0.0001 and a batch size of 24. However, this batchsize was only possible on an Nvidia Quadro GV100 with 32GB of video memory. Even with this powerful graphics card, the training time of this model was approximately 13 days.

**DenseNet** Densely Connected Convolutional Networks (Huang et al., 2017) are state-of-the-art for various image processing tasks. The basic idea of DenseNet is to re-use features learned on early layers of a network, containing fine-grained localized information, on higher layers which have a more abstract representation of the input. This is achieved by passing feature maps of a layer to all consecutive layers (within certain boundaries). A stated benefit of this architecture is that it does not have to re-learn features several times throughout the network. Hence, the individual convolutional layers have a relatively small number of learned filters. We used the variant DenseNet-121 with a learning rate of 0.01, SGD optimizer with momentum of 0.9, L2 weight decay of 0.0001 and a batch size of 12.

**Baseline Fully Connected Network (FNN)** In Simm et al. (2018), the best performing method was the fully-connected deep multi-task neural network. We re-implemented this architecture together with the best hyperparameters and used it as a baseline model to compare the convolutional networks against. We used parameters given by the authors, without dropout schedule, which improved the results.

**GapNet** While abstract features from deep convolutional layers are semantically strong, they lack spatial information necessary for detecting or taking into account smaller objects or input features. In Lin et al. (2017) the authors leverage features from multiple scale levels of the convolutional network by combining feature maps via scaling and subsequent 1×1 convolution while Godinez et al. (2017) process the input at different scales and combine the resulting feature maps late in the network. In Huang et al. (2017), feature maps from all scale levels are carried over to subsequent levels within so-called Dense-Blocks and thus can be re-used by higher layers of the network.

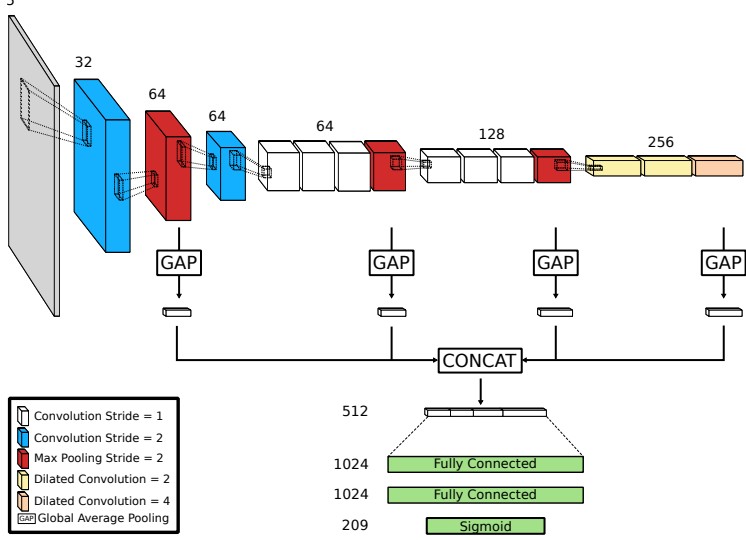

Figure 2: Schematic representation of the GapNet architecture. A standard CNN architecture with a sequence of 2D convolutions and max pooling is combined with global average pooling operations of particular feature maps. The resulting feature vectors are concatenated and fed into two fully-connected layers and an output layer.

We introduce a novel convolutional neural network architecture designed for the unique challenges of analyzing microscopy images for drug design. Since we make heavy use of *Global Average Pooling* we refer to it as *GapNet*. It consists of an encoder part using convolutional and pooling layers in a way that allows it to process high-resolution images efficiently while not losing information

due to downscaling operations in a preprocessing step. Additionally, we use *dilated convolutions* in the deepest layers of the encoder network tuned for a receptive field roughly equal to the dimensions of the input to the network to enable those layers to gather global information which has shown to be beneficial for some tasks. Features extracted by this convolutional encoder at different scales are then reduced via averaging over the individual feature maps, in effect producing feature statistics from different layers of abstraction and spatial resolution. These pooled features are then concatenated and processed by several fully-connected layers. Figure 2 shows the architecture with the parameters of the variant used for our experiments. We use the SELU (Klambauer et al., 2017) activation function throughout the network except for the output layer where we use the *sigmoid* activation function. As for hyperparameters, we used a learning rate of 0.01, SGD optimizer with momentum of 0.9, L2 weight decay of 0.0001 and a batch size of 128.

**Single-Cell CNN (SC-CNN)**    We included a convolutional neural network operating on small crops centered on single cells, SC-CNN. This method depends on a segmentation algorithm that had already identified the cell centers (Bray et al., 2017). We extracted crops of size $96 \times 96$ centered on individual cells, whose coordinates are taken from this pre-computed cell-segmentation included in the Cellpaining dataset. This crop-size was chosen such that the majority of cells are contained within a crop regardless of orientation. Due to this greatly reduced input size we modified the Gap-Net architecture slightly such that the dilated convolutions in the last block of the encoder have been replaced by regular convolutions as the large receptive field is detrimental for such small images. We used the same dataset split as for all other methods, resulting in 12.7 million crops for training, with 1.8 million and 3.6 million crops in the validation- and test-sets respectively. During training of the cell-centric model we randomly sampled a subset of 20% from the training-set each epoch. We used a learning rate of 0.01, SGD optimizer with momentum of 0.2, L2 weight decay of 0.0001 and a batch size of 2048.

### 3.1 EVALUATION CRITERIA

We used the area under ROC curve (AUC) as main evaluation criterion. This is the most relevant criterion in drug discovery, since compounds are selected from a ranked list for subsequent lab tests. The AUC was calculated per task, such that each method is characterized by 209 performance values across 209 prediction tasks. The difference between two methods is tested by a paired Wilcoxon test across these 209 AUC values, where the null hypothesis is that the two methods have equal performance.

## 4 RESULTS

| Model | Type | AUC | F1 | AUC >0.9 | AUC >0.8 | AUC >0.7 |
|-------|------|-----|----|----------|----------|----------|
| MIL-Net | end-to-end | **0.726±0.20** | 0.485±0.36 | 66 | 86 | 109 |
| ResNet | end-to-end | **0.722±0.21** | 0.516±0.32 | 69 | 88 | 119 |
| GapNet | end-to-end | **0.721±0.22** | 0.532±0.32 | 61 | 95 | 116 |
| DenseNet | end-to-end | **0.718±0.22** | 0.536±0.33 | 60 | 100 | 119 |
| M-CNN | end-to-end | 0.709±0.21 | 0.482±0.32 | 57 | 80 | 108 |
| SC-CNN | cell-centric | 0.695±0.22 | 0.371±0.30 | 57 | 82 | 109 |
| FNN | cell-centric | 0.677±0.22 | 0.372±0.33 | 57 | 72 | 90 |

Table 1: Model performances in terms of different performance metrics: mean AUC, mean F1 score and number of tasks (assays) that can be predicted with an AUC better than 0.9, 0.8, and 0.7. The columns AUC and F1 report the average AUC and F1 and its standard deviation across the 209 prediction tasks. CNNs operating on full high-resolution images significantly outperform SC-CNN which operates on cell-centric crops and the FNN operating on pre-computed features. End-to-end CNN models perform on par. Performance values marked in bold indicate that the best performing method does not significantly outperform the respective method.

**Method comparison across all assays**    The compared methods yielded mean AUCs from 0.677(±0.22) for FNNs to 0.726(±0.20) for MIL-Net, see Table 1 and mean F1 scores ranging

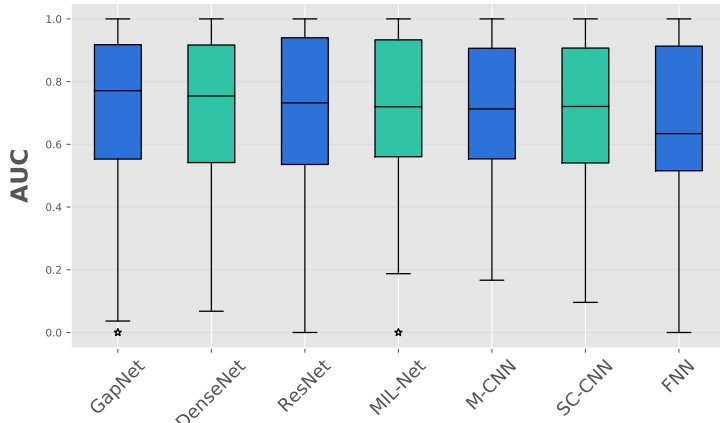

Figure 3: Boxplot comparing the performance of all methods. The x-axis displays the compared methods and the y-axis displays the performance across the 209 different assays as boxplots. GapNet exhibits the best median AUC value, whereas FNN exhibits the worst.

from 0.371±0.30 for SC-CNN to 0.536±0.33 for DenseNet. We report not only AUCs as assays are typically heavily imbalanced, thus the AUC may overestimate performance. All end-to-end convolutional networks performed significantly better than cell-centric methods while no CNN architecture significantly outperformed all other CNN architectures in predictive performance (see Figure 3 and Appendix Table A1). However, for the end-to-end approaches, GapNet exhibits the lowest training time and MIL-Net has the lowest number of parameters (see Table 2).

| Architecture | Parameters | Time per Epoch |
|---|---|---|
| GapNet | 3,694,545 | 662.04 |
| MIL-Net | 726,180 | 1344.18 |
| M-CNN | 11,230,929 | 4252.47 |
| DenseNet | 7,174,353 | 8519.91 |
| ResNet | 42,934,673 | 13208.64 |

Table 2: Number of Parameters and runtime (in seconds per epoch) of the end-to-end CNN architectures on an Nvidia GTX 1080 TI.

**Method comparison at assay level**  We also investigated whether the difference in predictive performance of the end-to-end approach of GapNet and the cell-centric approach using FNNs is related to the type of the modeled assay. Overall, we found that the predictive performance of these two approaches is highly correlated across assays (Pearson correlation of 0.72, see Figure 4). Oftentimes, there is no significant difference in performance (Venkatraman's test for difference in AUC curves (Venkatraman, 2000)). However, for 29 assays, GapNet is significantly better than the FNN, indicating that CNNs detect morphological characteristics that are not captured by the pre-computed features. Furthermore, while for cell-centric activity prediction the number of measured compounds strongly influences performance, the trend is less pronounced for image-based activity prediction (see Appendix Figure A1).

## 5  DISCUSSION

We investigated the potential of end-to-end learning approaches on high-resolution microscopy imaging data to predict pharmacological assays. Despite the common opinion that these data require a cell-centric approach involving a segmentation algorithm, we found that CNNs learned in an end-to-end fashion not only are feasible but even outperform cell-centric approaches. Our results indicate that CNNs are able to extract better features from images as every end-to-end CNN model in our comparison outperformed the methods based on cell-centric crops. Our results demonstrate

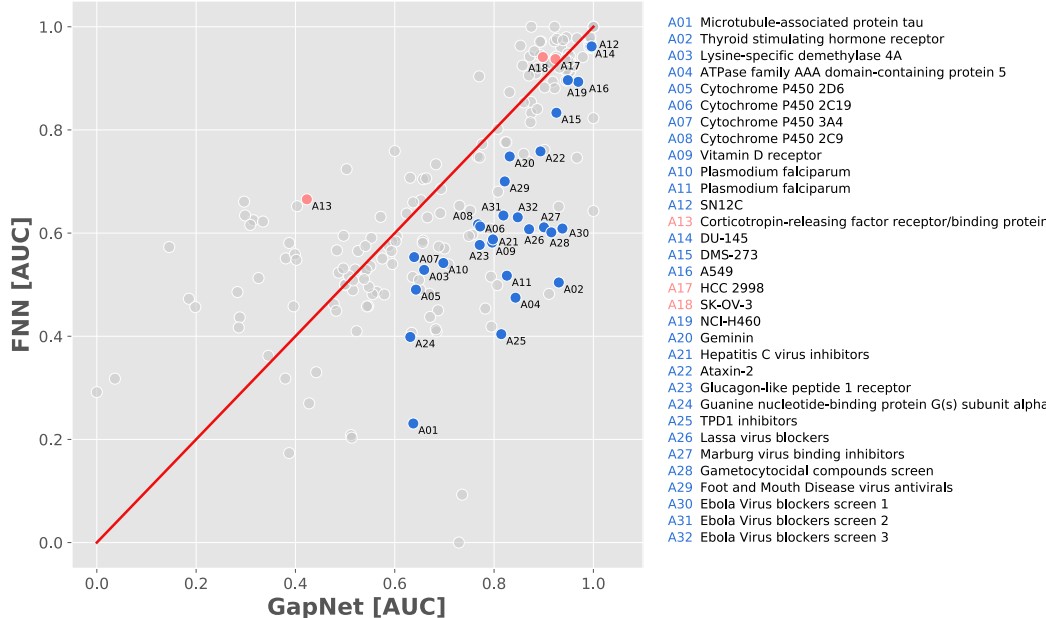

Figure 4: Comparison of FNN performance with GapNet over all tasks. The x-axis displays the performance in terms of AUC of GapNet at the 209 assays, whereas the y-axis shows the performance of FNN. For a large number of assays, there is no significant difference in predictive performance (grey dots), whereas for 29 assays, GapNet significantly outperformed FNN (blue dots) and for 3 assays FNN significantly outperformed GapNet (red dots). Overall, predictive performance of the two compared methods is highly correlated across assays, which indicates that if a biological effect expresses in morphological changes in the cells, both approaches capture it to a certain degree.

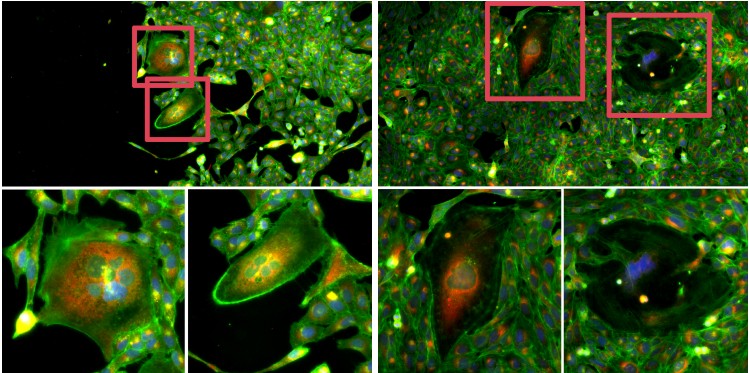

Figure 5: Examples from the assay "Gametocytocidal compounds screen" treated with compounds predicted and labeled as active. While cell density is not significantly different from untreated samples, cells show distinct morphological changes. These changes can hamper segmentation and feature extraction algorithms, thus ignoring these indicative cells.

that the complicated and costly cell segmentation and feature extraction step is not necessary, but rather should be skipped to obtain better predictive performance.

With CNNs and high-resolution fluorescence imaging, many new relations between cell morphology and biological effects could be detected and used to annotate chemical compounds at low cost. Figure 5 shows examples treated with compounds predicted as active from the assay "Gametocytocidal compounds screen" (see A28 in Figure 4) where the CNN performs significantly better than the FNN. Here we see, that while overall cell density is not significantly different from samples treated with inactive compounds some cells show clear morphological changes which might be indicative

for the classification of the compound as *active* in the respective assay. Such drastic changes can be a major problem for segmentation and feature extraction algorithms, which might result in missed detection of these features. Even if these few abnormal cells are detected, their signal may be lost when averaging across cells (Simm et al., 2018). We hypothesize that the increased performance of CNNs arises from problems of cell segmentation and detection of sparse signals.

The fact that the Cellpainting assay protocol has been published and similar images could be produced in many labs across the world, opens the opportunity that our trained network could be used to automatically annotate these images. With the currently available data, we were able to annotate $\approx$30,000 compounds in 61 assays at high predictive performance (AUC>0.9), which amounts to approximately 1.8 million lab tests. We envision that this work could make world-wide drug discovery efforts faster and cheaper.

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

## 6 APPENDIX

### 6.1 WILCOXON TEST

|          | MIL-Net   | ResNet    | GapNet   | DenseNet | M-CNN    | SC-CNN  |
|----------|-----------|-----------|----------|----------|----------|---------|
| ResNet   | 3.50e-01  |           |          |          |          |         |
| GapNet   | 2.49e-01  | 5.34e-01  |          |          |          |         |
| DenseNet | 1.61e-01  | 1.52e-01  | 2.09e-01 |          |          |         |
| M-CNN    | **3.75e-04** | **7.18e-03** | 4.07e-02 | 1.54e-01 |          |         |
| SC-CNN   | **2.02e-07** | **5.32e-06** | **4.98e-06** | **6.98e-04** | 1.65e-02 |         |
| FNN      | **7.67e-12** | **7.11e-07** | **2.40e-05** | **6.29e-04** | **1.08e-06** | 1.61e-02 |

Table A1: p-values of paired Wilcoxon test with the alternative hypothesis that the column method has outperformed the row method. Significant values with $\alpha = 0.01$ are marked in bold.

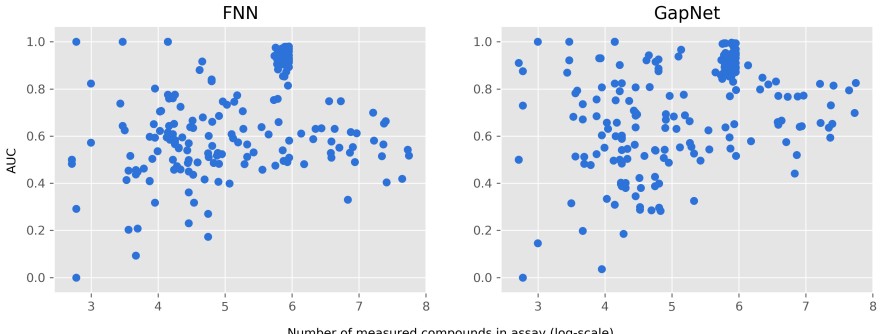

Figure A1: Relation between the number of measured compounds for each assay and the predictive performance of the FNN (left) and GapNet (right). For the FNN approach the number of measured compounds strongly influences performance while this thrend is less pronounced for the CNN based end-to-end approach. Other end-to-end approaches look similar to GapNet.

