# OpenReview forum: "End-to-end learning of pharmacological assays from high-resolution microscopy images"
_ICLR.cc/2019/Conference_

### Official Review · AnonReviewer2 · 2018-11-02
**A new and interesting application but the strength of original contributions is unclear**

**Rating:** 5
**Confidence:** 4

**Review:**

The authors explore the possibility of using an end-to-end approach for predicting pharmacological assay outcome using fluorescence microscopy images from the public Cell Painting dataset. In my view, the primary contributions are the following: an interesting and relatively new application (predicting assay outcomes), enriching the CellPainting dataset with drug activity data, and a comparison of several relevant methods and architectures. The technical novelty is weak, and although the authors demonstrate that end-to-end holistic approaches outperform previous segmentation-and-feature-extraction approaches, this result is not surprising and has been previously reported in closely related contexts.


OVERVIEW

The authors evaluate the possibility of using and end-to-end deep learning approach to predict drug activity using only image data as input. The authors repurpose the CellPainting dataset for activity prediction by adding activity data from online ChEMBL databases. If made available as promised, the dataset will be a valuable resource to the community. The authors compare a number of previous approaches and state-of-the-art image classification network architectures to evaluate the use of CNNs instead of more classical image analysis pipelines. The comparison is a strong point of the paper, although some details are lacking. For example, the authors claim that GapNet is the quickest method to train, and while they report the number of hyperparameters and time per epoch, the number of epochs trained is never mentioned.

The authors propose an architecture (GapNet) for the assay prediction task. While the way Global Average Pooling is used to extract features at different stages in the network might be new, it is a straightforward combination of GAP and skip connections. Little insight into why this approach is more efficient or evidence for its effectiveness is provided. Similarly, more explanation for why dilated convolutions and SELU activations would be appreciated. A comparison between GapNet and the same network without the GAP connections could possibly provide a more interesting comparison and might also provide a more pervasive argument as to why GapNet’s should be used. Ultimately, the benefit of using GapNet over the other architectures is not strongly motivated, as training time is less of a concern in this application than predictive power.


RELATED WORK

The authors present previous work in a clear and comprehensive manner. However, the reported finding that “CNNs operating on full images containing hundreds of cells can perform significantly better at assay prediction than networks operating on a single-cell level” is not surprising, and partial evidence of this can be found in the literature. In [1], it was shown that penultimate feature activations from pre-trained CNNs applied to whole-image fluorescence microscopy data (MOA prediction) outperform the baseline segmentation-then-feature extraction method (FNN). Similarly, in [2] (the paper proposing MIL-Net), it is shown that end-to-end whole-image CNN learning for protein localization outperforms the baseline (FNN). In [3] whole image end-to-end learning outperforms whole image extracted features for a phenotyping task. All of these references use fluorescence microscopy data similar to the dataset in this work.

[1] Pawlowski, Nick, et al. "Automating morphological profiling with generic deep convolutional networks." bioRxiv (2016): 085118.
[2] Kraus, Oren Z., Jimmy Lei Ba, and Brendan J. Frey. "Classifying and segmenting microscopy images with deep multiple instance learning." Bioinformatics 32.12 (2016): i52-i59
[3] Godinez, William J., et al. "A multi-scale convolutional neural network for phenotyping high-content cellular images." Bioinformatics 33.13 (2017): 2010-2019.


APPROACH

The authors compile enrich the CellPaining dataset with activity data from various drug discovery assays. In my view, the creation of this dataset is the strongest and most valuable contribution of the paper. The method used to collect the data is described clearly and the choices made when compiling the dataset, including the thresholds and combinations of activity measures seems like a well founded approach.

The authors then identify a number of approaches that are relevant for the problem at hand, binary prediction of drug activity based on image data. These include previous approaches used for cell images and modern image classification networks.


EXPERIMENTS

The different approaches/networks mentioned above were evaluated on a testset. The results indicate that end-to-end CNN approaches outperform all non-end-to-end with no significant difference between the individual end-to-end CNNs. The results are stated clearly and the presentation of different metrics is a nice addition to properly compare the results. It would however contribute valuable information if the authors stated how the confidence intervals of the F1 score are calculated (are the experiments based on several runs of each network or how is it done).


NOVELTY/IMPACT

+ Creation of a new dataset on a new and interesting problem
+ Useful comparison of modern networks on the task
- GapNet - lacking technical novelty, insight, and performance is unconvincing
- Demonstrates that end-to-end learning outperforms cell centric approach - was this really surprising or even new information?


OTHER NOTES:
* Figure 3 is never mentioned in the main text
* Figure 3 (*’s) are confusing. Do they represent outliers? Statistical significance tests?
* Figure 5 which panel is which?
* Be clear what you mean when you refer to “upper layers” of a network
* An important point not mentioned: in practice, many assays use stains that are closely tied to the readout, unlike the dataset here which provides only landmark stains. The results found here do not necessarily apply in other cases.

---

> ### Author Response · Authors · 2018-11-26
> **Response to reviewer**
>
> We thank the reviewer for this assessment of our work.
>
> Indeed, we believe that our contribution lies in the assembly of a novel and
> relevant prediction task, and that we showed that end-to-end learning outperforms other
> approaches at this task. Although the machine learning community might have
> commonly hypothesized that the superiority of end-to-end learning would
> also hold in this area, our empirical evaluation is the first to demonstrate this.
> We also suggest a much more compact architecture, GapNet, that given a fixed time-span
> allows to search many more hyperparameters compared to other architectures.
>
>
> OTHER NOTES:
>
> 1) Ad Table 1
> The error bars in Table 1 are standard deviations of the AUC and F1 scores across prediction tasks.
> We apologize for not making this clearer and we now state this clearer in the table caption.
>
> 2) Ad Figure 3
> Thank you for pointing out the missing reference, we corrected this mistake in the manuscript.
> Also, the *’s represent outliers in this box plot.
>
> 3) Ad Figure 5
> Figure 5 shows example associated with active compounds according to a specific assay.
> The two bigger images on top are individual samples while the two smaller image below
> each of these shows the enlarged crop marked with red rectangles.
>
> 4) Be clear what you mean when you refer to “upper layers” of a network
> Thank you for pointing this out, we changed the wording to “deepest layers” which makes this more clear.
>
> 5) Relevance to other cases
>
> We thank the reviewer for this valuable comment that we had not addressed.
> Indeed, there are many assays, such as the micronucleus test, that are actual imaging readouts.
> However, the assays that we modelled here are cell-based assays whose association with
> this particular cell line and stains had not been investigated.

---

### Official Review · AnonReviewer1 · 2018-11-02
**An empirical study with little analysis**

**Rating:** 3
**Confidence:** 3

**Review:**

Edit: changed "Clarity"

[Relevance] Is this paper relevant to the ICLR audience? yes

[Significance] Are the results significant? no

[Novelty] Are the problems or approaches novel? no

[Soundness] Is the paper technically sound? okay

[Evaluation] Are claims well-supported by theoretical analysis or experimental results? marginal

[Clarity] Is the paper well-organized and clearly written? no

Confidence: 3/5

Seen submission posted elsewhere: No

Detailed comments:

In this work, the authors compare several state-of-the-art approaches for high-resolution microscopy analysis to predicting coarse labels for the outcomes of pharmacological assays. They also propose a new convolutional architecture for the same problem. An empirical comparison on a large dataset suggests that end-to-end systems outperform those which first perform a cell segmentation step; the predictive performance (AUC) of almost all the end-to-end systems is statistically indistinguishable.

=== Major comments

The paper is primarily written as though its main contribution is as an empirical evaluation of different microscopy analysis approaches. Recently, there have been a large number of proposed approaches, and I believe a neutral evaluation of these approaches on datasets other than those used by the respective authors would be a meaningful contribution. However, the current paper has two major shortcomings that prevent it from fulfilling such a place.

First, the authors propose a novel approach and include it in the evaluation. This undercuts claims of neutrality. (Minor comments about the proposed approach are given below.)

Second, the discussion of the results of the empirical evaluation is restricted almost solely to repeating in text the what the tables already show. Further, the discussion focuses only on the “top line” numbers, with the exception of a deep look at the Gametocytocidal compounds screen. It would be helpful to instead (or additionally) identify meaningful trends, supported by the data acquired during the experiments. For example: (1) Do the end-to-end systems perform well on the same assays? (2) Would a simple ensemble approach improve things? if they perform well on different assays, then that suggests it might. (3) What are the characteristics of the assays on which the CNN-based approaches perform well or poorly (i.e., how representative is Figure 5)? (4) What happens when the FNN-based approach outperforms the CNN-based ones? in particular, what happens in A13? (5) How sensitive are the approaches to the number of labeled examples of each assay type? (6) Are there particular compounds which seem particularly informative for different assays?

A second major concern is whether the binarized version of this problem (i.e., assay result prediction) is of interest to practitioners. In many contexts, quantitative information is also important (“how much of a response do we see?”). While one could imagine the rough qualitative predictions (“do we see a response?”) shown here as an initial filtering step, it is hard to believe that the approach proposed here would replace other more informative analysis approaches.

=== Minor comments

Are individual images from the same sample image always in only the training, validation, or testing set? that is, are there cases where some of the individual images from a particular sample image are in the training set, while others from that sample image are in the testing set?

I did not find the dataset construction description very clear. Does each row in the final, 10 574 x 209 matrix correspond to a single image? Does each image correspond to a single row? For example, it seems as though multiple rows may correspond to the same image (up to four? the three pChEMBL thresholds as well as the activity comment). What is the order in which the filtering and augmenting happens? It would be very helpful to provide a coherent, pipeline description of this (say, in an appendix).

Do all the images in the dataset come from the same microscope (and cell line) at the same resolution, zoom, etc.? If so, it is unclear how well this approach may work for images which are more heterogeneous. There are not very many datasets of the size described (I believe, at least) available. This may significantly limit the practical impact of this work.

How many epochs are required for convergence of the different architectures? For example, MIL-net has significantly fewer parameters than the others; does it converge on the validation set faster?

=== Typos, etc.

The references are not consistently formatted.

“not loosing” -> “not losing”
“doesn’t” -> “does not”

---

> ### Author Response · Authors · 2018-11-26
> **Response to reviewer**
>
> We thank the reviewer for his insightful comments and questions.
>
> === Major comments
> We agree with the reviewer that this paper empirically evaluates different microscopy analysis approaches.
> Indeed, our custom-designed architecture “GapNet” has been included in the analysis, but does not undercut neutrality.
> All architectures were adapted to this task, had the chance to adjust their most important hyperparameters
> on a validation set, and have been compared on an independent test set.
> We apologize that the discussion section does not meet the expectations of the reviewer.
>
> Regarding a more in-depth discussion of results:
>
> (1) Do the end-to-end systems perform well on the same assays?
> Yes, there is an overlap of 78%-89% between model on assays with AUC >0.9.
>
> (2) Would ensemble improve performance?
>
> As the reviewer points out, an ensemble of multiple models might increase predictive performance given the
> high overlap across highly-predictive assays for end-to-end models.
> Unfortunately we did not have the capacity to learn ensemble approaches at this time.
>
> (3) How representative is Figure 5?
>
> The reviewer points out a very interesting question here. We are looking into this by using
> contribution analysis (e.g. Integrated Gradients) to identify visual cues indicating bioactivity.
> However, due to the large number of samples and their complex nature this is still an ongoing process.
>
> (4) Discussion of FNN outperforming CNN
>
> Similar to the previous question we are still in the process of gaining insights here.
> With respect to this specific assay it seems that the CNN-based method is unable to predict any
> sample correctly as active (although it is capable of correctly predicting inactive samples).
> From a first visual inspection we are unsure what causes this.
>
> (5) Sensitivity with respect to the number of labeled examples per assay
>
> We checked the relation of number of labeled examples per assay with its AUC for all models and overall there is only a slight positive trend. Both end-to-end models as well as the FNN-based approach are able to achieve high AUCs for assays with a high number of labeled samples (200+ and more) as well as for those with a lower number of labeled samples (starting with ~50 samples we see AUCs of >0.9). We added a figure that displays this association (Appendix Figure A1).
>
> (6) Informative compounds and binarized labels
>
> We agree with the reviewer that modeling this as a regression task could provide valuable insights
> or be of more interest to practitioners.
> However, we aim at being comparable to the work of Simm et al. (2018) who also considered
> this as a classification task. Nonetheless, we plan to include modelling of the regression
> task in a future version of this work.
>
>
> === Minor comments
> 1) Are individual images from the same sample image always in only the training, validation, or testing set?
>
> Yes, we took care that multiple images from the same sample are always in the same fold.
> We now state this clearer in the manuscript.
>
> 2) I did not find the dataset construction description very clear.
>
> We apologize for not stating the process clearly and have improved the description of the
> data set which hopefully makes this more clear.
>
> 3) Pipeline and data collection
>
> We have improved the pipeline description in the paper.
> Regarding dataset collection, yes all images have been captured with the same microscope and magnification and are of the same cell line (U2OS).
> Due to the absence of a data set comprising images from multiple devices and labs, it is yet unclear how robust the predictive performance is with respect to variance arising from those sources.
>
> 4) Convergence of different architectures
>
> Actually, MIL-Net takes the most epochs to converge on the validation set, only reaching good performance after about 90 epochs. In contrast GAP-Net, DenseNet and ResNet converge somewhere around 50-60 epochs. M-CNN converges after roughly 40 epochs and the FNN after 65 epochs.

---

> > ### Comment · AnonReviewer1 · 2018-11-26
> > **Response does not change view**
> >
> > I appreciate the authors' responses to my comments; however, they do not really address my concerns about the contribution of the empirical comparison. I believe a revised version of the paper which addresses some of the questions which are still open (in particular, 3 and 4) would significantly improve the contribution.

---

### Official Review · AnonReviewer3 · 2018-11-02
**The paper introduces Gapnet, that uses a CNN architecture to learn pharmacological assays from high-resolution microscopy images. The paper deals with a valid problem of handling images in a segmentation-agnostic way.**

**Rating:** 6
**Confidence:** 5

**Review:**

The paper is well written, deals with a valid and crucial end-to-end imaging problem.

Comments
1) Section 2: It is not clear how 10574 compounds increase to 11585 (2nd paragraph page 3). Also how does one arrive at 11171 compounds (para 3).
2) How do you arrive at 209 assays from 10818?
Do consider enumerating this Section: data dimensions you started with and then how the dimensions were reduced per step. I gather you have mentioned this but it is confusing to grasp, at this point.

3) In page 2, you mention the images have 5 channels but towards the end of the section on page 3, it says 1) views have ‘6’ such images per sample image and 2) 4 channels for stains. How many stains are there per channel and how are 5 channels related to the ‘6’ and 4 channels?

4) In Section 4 and Appendix 6, it does not seem that Gapnet outperforms, rather it is at par to, other architectures. Is the only gain with Gapnet the runtime across epochs?

---

> ### Author Response · Authors · 2018-11-26
> **Response to reviewer**
>
> We thank the reviewer for his/her positive comment on our manuscript.
>
> Regarding questions 1) and 2) we apologize for the confusing explanation of the process to generate our label matrix, we have tried to state the process more clearly in the paper.
>
> The final label matrix has 10,057 compounds and 209 assays. To arrive at that we perform the following steps:
>   1) Starting with the full ChEMBL database, extract all compounds for which we have microscopy images (11,585)
>   2) Extract all assays for which at least one compound with a pChEMBL value between 4 and 10 is present.
>   3) Then, we apply three thresholds to this matrix (5.5, 6.5 and 7.5) with values above the threshold indicating an
>        active compound and below an inactive compound.
>   4) These three binary matrices are concatenated along the assay dimension.
>   5) Next, we also use the “activity comment” field of ChEMBL which directly gives us a binary matrix.
>   6) We concatenate the three thresholded matrices and the activity comment matrix.
>   7) Finally, we keep only assays (or assay/threshold combinations) with at least 10 active and 10 inactive compounds.
>   8) This results in some compounds now having no measurement in the remaining assays and by removing these we
>        arrive at 10,574 compounds and 209 assays.
>
> Regarding question 3), the 4 channels stated on page 3 were actually a typo - we now correctly state 5 channels. We apologize for this error and the the unclear explanation of channels, views, and images.
> The microscopic device takes six adjacent images, called “views”, for each sample. These six views can be stitched together to get one image per sample (they are arranged in a 2x3 grid). Each of these images, or views, has five channels that correspond to five different fluorescent dyes. We improved this description in the main manuscript.
>
> Ad question 4:
> As the reviewer correctly pointed out, GapNet is on par with other architectures w.r.t. Predictive performance, but fast w.r.t. computation speed.

---

### Meta-Review · Area_Chair1 · 2018-12-12
**Interesting study applying CNNs to prediction of assays, but work is perhaps more suited for a biomedical imaging journal.**

**Confidence:** 5
**Recommendation:** Reject

**Metareview:**

This work studies the performance of several end-to-end CNN architectures for the prediction of biomedical assays in microscopy images. One of the architectures, GAPnet, is a minor modification of existing global average pooling (GAP) networks, involving skip connections and concatenations. The technical novelties are low, as outlined by several reviewers and confirmed by the authors, as most of the value of the work lies in the empirical evaluation of existing methods, or minor variants thereof.

Given the low technical novelty and reviewer consensus, recommend reject, however area chair recognizes that the discovered utility may be of value for the biomedical community. Authors are encouraged to use reviewer feedback to improve the work, and submit to a biomedical imaging venue for dissemination to the appropriate communities.